# Effect and Mechanism of L-Arginine against *Alternaria* Fruit Rot in Postharvest Blueberry Fruit

**DOI:** 10.3390/plants13081058

**Published:** 2024-04-09

**Authors:** Jiaqi Wang, Runan Zhao, Yuxuan Li, Haifeng Rong, Ling Yang, Ming Gao, Bingxin Sun, Yunhe Zhang, Yufeng Xu, Xuerui Yan

**Affiliations:** 1College of Food Science, Shenyang Agricultural University, Shenyang 110866, China; jiaqiwang@stu.syau.edu.cn (J.W.); 2020187023@stu.syau.edu.cn (R.Z.); 2021200019@stu.syau.edu.cn (Y.L.); rhf888@stu.syau.edu.cn (H.R.); 17624063742@163.com (L.Y.); 2022220091@stu.syau.edu.cn (M.G.); packsun@163.com (B.S.); 2018500021@syau.edu.cn (Y.Z.); xuyf@syau.edu.cn (Y.X.); 2Key Laboratory of Protected Horticulture (Shenyang Agricultural University), Ministry of Education, Shenyang 110866, China; 3Shenyang Key Laboratory for Logistics Preservation and Packaging of Agricultural Products, Shenyang 110866, China

**Keywords:** L-arginine, blueberry fruit, reactive oxygen species, jasmonic acid biosynthesis, pathogenesis-related proteins

## Abstract

This study aimed to explore the impact of L-arginine (Arg) on the development of resistance to *Alternaria tenuissima* (*A. tenuissima*) in blueberries. The metabolism of reactive oxygen species, pathogenesis-related proteins (PRs), and jasmonic acid (JA) biosynthesis pathways were analyzed, including changes in activity and gene expression of key enzymes. The results indicated that Arg treatment could prevent the development of *Alternaria* fruit rot in postharvest blueberries. In addition, it was also found to induce a burst of hydrogen peroxide in the blueberries early on during storage, thereby improving their resistance to *A. tenuissima*. Arg treatment was observed to increase the activity of antioxidant enzymes (peroxidase, catalase, superoxide dismutase, and ascorbate peroxidase) and related gene expression, as well as the total levels of phenolics, flavonoids, and anthocyanin in the blueberries. The activity and gene expression of the PRs (chitinase and β-1,3-glucanase) were elevated in Arg-treated blueberries, boosting their resistance to pathogens. Additionally, a surge in endogenous JA content was detected in Arg-treated blueberries, along with upregulated expression of key genes related the JA biosynthesis pathway (*VcLOX1*, *VcAOS1*, *VcAOC*, *VcAOC3*, *VcOPR1*, *VcOPR3*, *VcMYC2*, and *VcCOI1*), thereby further bolstering disease resistance. In conclusion, Arg treatment was determined to be a promising prospective method for controlling *Alternaria* fruit rot in blueberries.

## 1. Introduction

Blueberries (*Vaccinium* spp.) are highly perishable due to their soft skin, which leaves them susceptible to softening and rotting, leading to diminishing taste and nutritional value [1,2]. Blueberries are prone to mechanical damage after harvest and are easily infected by pathogens, such as *Botrytis cinerea* and *Alternaria* sp. These pathogens are known to cause fruit rot in postharvest blueberries [3,4]. Zhu and Xiao isolated and identified 283 strains of *Alternaria* sp. from rotten blueberry fruit in California, USA, including five species, *Alternaria alternata*, *A. tenuissima*, *A. arborescens*, *A. infectoria*, and *A. rosae*, respectively [5]. In our previous study, we have isolated and identified *A. alternata*, *A. tenuissima*, and *A. dumosa* in rotten blueberry fruit in Liaoning, China. Munitz et al. have found that *A. tenuissima* was the dominant pathogen of blueberry fruit in Concordia, Entre Ríos Province, Argentina [6]. In recent years, controlling *Alternaria* fruit rot in blueberries has attracted attention. Fungicides such as natamycin have been reported to control *Alternaria* fruit rot in blueberries [7]. Ethanol vapor has been found to enable effective defense responses against *Alternaria* rot in blueberries [8]. Biological control has also been used to prevent *Alternaria* fruit rot, such as the use of *Bacillus subtilis* (F9-2 and F9-12) and *Pseudomonas koreensis* (F9-9) [9]. For the past few years, several studies have been conducted to test the ability of natural compounds to control fruit rot in fruits and vegetables during postharvest. Of these, L-arginine (Arg) has gained interest among researchers.

Arg belongs to a group of semi-essential amino acids that are safe and beneficial for human health [10]. There has been a growing interest in the potential functions of Arg related to plants’ stress responses. In previous studies, tomato fruit treated with Arg and methyl salicylate has been found to effectively resist rot caused by *Botrytis cinerea* [11]. Arg treatment was also found to induce resistance to *Alternaria alternata* in jujube fruit [12]. Our previous study found that Arg treatment maintained the quality of postharvest blueberries and activated the antioxidant system [13]. However, the effect and potential mechanism of Arg treatment in inducing resistance to *A. tenuissima* in blueberries remains unclear.

Plants have specific defense responses, such as the production of reactive oxygen species (ROS) and pathogenesis-related proteins (PRs), in the face of stress triggers [14]. In addition, jasmonic acid (JA) is a kind of endogenous plant hormone that is important for defense against pathogens in plants [15]. JA has been found to trigger a generation of secondary metabolites that induce systemic resistance (ISR) in plants, which can not only give rise to physiological changes that form defensive structures but also induce gene expression [16]. ISR mediated by JA, along with JA accumulation, is related to the regulation of resistance to necrotrophs in plants [17]. In the JA biosynthesis pathway, key enzymes such as lipoxygenase (LOX), 12-oxo-phytodienoic acid reductase (OPR), allene oxide cyclase (AOC), and allene oxide synthase (AOS), play important roles [18,19,20]. 

Through our previous research, we determined that Arg treatment maintains fruit quality in postharvest blueberries by improving their antioxidant capacity. However, it is still unclear whether Arg treatment can induce resistance to *A. tenuissima*-related decay in postharvest blueberries. Hence, this study aims to explore whether Arg treatment can induce postharvest resistance in blueberries contaminated by *A. tenuissima*. Through this study and the analysis of the activities and related gene expression levels of ROS metabolism and PRs, we aim to elucidate the mechanism of defense response initiation in Arg-treated blueberries. We also aim to monitor endogenous JA content and key gene expression levels related to metabolic pathways after Arg treatment. Our findings could be meaningful for controlling postharvest disease in blueberries and may provide novel evidence supporting the use of Arg.

## 2. Results

### 2.1. Arg Treatment Mitigated Disease Symptoms in Blueberries Inoculated with A. tenuissima

The blueberries began to rot two days post inoculation (Figure 1A). Over time, the disease index was found to increase gradually in both groups. During storage, Arg treatment was found to significantly reduce the disease index in the blueberries (*p* < 0.05). At six days, the disease index in the Arg-treated group was 32.05% lower than in the control. The fruit rot symptoms at six days are shown in Figure 1B. These findings indicate that Arg treatment inhibited the blueberry fruit rot caused by *A. tenuissima*.

### 2.2. Effect of Arg Treatment on β-1,3-Glucanase (GLU) and Chitinase (CHI) Activities and Relative Gene Expression Levels in Inoculated Blueberries 

It was observed that GLU activity in Arg-treated blueberries increased progressively and peaked at three days, before steadily declining (Figure 2A). The control blueberries displayed a similar pattern, with GLU activity peaking at four days. The GLU activity was found to be significantly higher in Arg-treated blueberries than control at 0.25, 0.5, 1, 2, and 3 days (*p* < 0.05). Furthermore, the CHI activity of Arg-treated fruit was significantly higher than the control group, at 0.25, 2, 4, 5, and 6 days (*p* < 0.05) (Figure 2C). 

We also measured the relative gene expressions of the involved enzymes. Throughout storage, Arg was found to increase *VcGLU2* expression, which remained significantly higher in the Arg-treated blueberries at 0.25–2 days as compared to the controls (Figure 2B). Notably, *VcCHI* expression increased within the first two days but decreased thereafter (Figure 2D). *VcCHI* expression was found to be significantly higher in Arg-treated blueberries at 0.5 days and 1 day than in the controls. 

### 2.3. Effect of Arg Treatment on Endogenous JA Content and Gene Expression Levels Involved in JA Biosynthesis in Inoculated Blueberries

According to Figure 3A, the JA content in the Arg-treated blueberries peaked at four days but subsequently decreased between four and six days. Meanwhile, the endogenous JA content of the Arg-treated blueberries increased before four days and remained significantly higher compared to that of the control blueberries at 1, 2, 3, and 4 days. It was found to be 2.33 times higher than the control at four days. These findings indicate that Arg treatment stimulated the synthesis of endogenous JA.

Moreover, we analyzed the relative gene expression levels involved in JA biosynthesis, containing *VcLOX1*, *VcAOS1*, *VcAOC*, *VcAOC3*, *VcOPR1*, *VcOPR3*, *VcMYC2*, and *VcCOI1* (Figure 3). As Figure 3B illustrates, *VcLOX1* expression remained significantly higher in Arg group than the control blueberries during the initial time of storage (0.25, 0.5, and 1 day). *VcAOS1* expression in the Arg-treated group rose sharply in two days, remaining significantly higher than the control at 0.5 and 2–4 days (Figure 3C). The expression of *VcAOS1* in Arg-treated blueberries peaked at four days, approximately 3.12 times higher than in the controls. *VcAOC* expression in Arg-treated blueberries remained significantly higher than in the controls at 0.25, 0.5, and 3–5 days (*p* < 0.05) (Figure 3D). As demonstrated in Figure 3E, *VcAOC3* expression was found to be significantly higher in the Arg treatment group at 1, 3, and 4 days (*p* < 0.05). The relative expression of *VcOPR1* peaked at four days and subsequently declined (Figure 3F). *VcOPR1* expression in the Arg-treated blueberries was observed to be significantly higher than in the control blueberries at 0.5–4 days. Nevertheless, the expression of *VcOPR3* in Arg treatment remained significantly higher than control at 0.5 days and 1 day (Figure 3G). As shown in Figure 3H, Arg treatment upregulated *VcMYC2* expression in blueberries. Notably, *VcMYC2* expression in Arg-treated blueberries peaked at two days, 4.20 times higher than in the controls. As demonstrated in Figure 3I, Arg-treated blueberries exhibited higher *VcCOI1* expression than the controls at 0.25 and 0.5 days (*p* < 0.05). Hence, it was indicated that *VcLOX1*, *VcAOS1*, *VcAOC*, *VcAOC3*, *VcOPR1*, *VcOPR3*, *VcMYC2*, and *VcCOI1* were substantially upregulated in Arg-treated blueberries. Overall, exogenous Arg treatment enhanced the endogenous JA content of the blueberries, likely by activating related genes in the JA biosynthesis pathway.

### 2.4. Effect of Arg Treatment on Hydrogen Peroxide (H_2_O_2_) and Malondialdehyde (MDA) Content in Inoculated Blueberries

Changes in H_2_O_2_ content after inoculating *A. tenuissima* in the blueberries were determined (Figure 4A). Variation in H_2_O_2_ content exhibited similar trends in both groups. However, H_2_O_2_ content in the Arg treatment group was higher at 0.5 days and 1 day, indicating that Arg treatment catalyzed an early occurrence of oxidative stress. During the middle and late stages (2, 3, 5, and 6 days), it was observed that the Arg-treated blueberries produced significantly less H_2_O_2_ than the control blueberries. MDA content was significantly lower in the Arg-treated blueberries at 2–6 days than in the controls (Figure 4B).

### 2.5. Effect of Arg Treatment on Antioxidant Enzyme Activities and Related Gene Expression in Inoculated Blueberries

Following Arg treatment, the activity of superoxide dismutase (SOD) rose during the first two days and declined thereafter (Figure 5A). Compared to the controls, the Arg-treated blueberries exhibited higher SOD activity during storage; the difference was significant except at two and five days. The catalase (CAT) and ascorbate peroxidase (APX) activity of the Arg group were higher than at 0 d. CAT activity in the Arg-treated blueberries rose before the initial 0.5 days, and was three-fold higher than that of the controls (Figure 5C). Moreover, Arg-treated blueberries showed significantly higher CAT activity than the controls at 0.5, 1, 4, and 5 days. APX activity peaked at one day in both groups (Figure 5E) and remained significantly higher in the Arg treatment group at 0.25, 0.5, 3, and 4 days than in the controls. Peroxidase (POD) activity in the Arg-treated blueberries was found to be significantly higher than in the controls at 0.25, 0.5, and 3 days (Figure 5G).

The expression levels of the antioxidant enzymes in blueberries inoculated with *A. tenuissima* were evaluated (Figure 5). *VcSODCC.2* expression in the Arg-treated group increased first and peaked at two days, significantly higher than control at initial time (0.25–2 days) (Figure 5B). As evinced in Figure 5D, the *VcCAT2* expression in the Arg group was significantly higher than in the controls at 0.5–3 days of storage. Likewise, Arg treatment was found to promote *VcAPX1* expression during storage (Figure 5F). The difference was significant at 1–3 d. However, there was a significant drop in *VcPOD24* expression observed at 0.5 days in control (Figure 5H). The Arg-treated blueberries exhibited significantly higher *VcPOD24* expression than control during storage at 0.5 and 4 days. Based on these findings, it is indicated that Arg treatment triggered an antioxidative response in blueberries infected by *A. tenuissima*.

### 2.6. Effect of Arg Treatment on Total Phenolic, Flavonoid, and Anthocyanin Content in Inoculated Blueberries

Total phenolic content in both groups remained steady even at prolonged storage times. At 0.25, 2, 3, and 4 days, total phenolic content in the Arg-treated blueberries was significantly higher than in the controls (Figure 6A). Notably, at day 2, the total phenolic content following Arg treatment was 26.9% higher than control. Moreover, the flavonoid content in the Arg group was significantly higher than control at 0.5 and 2–4 days, peaking at 4 days (Figure 6B). The anthocyanin content increased first and declined thereafter in both groups, with the Arg-treated blueberries exhibiting significantly higher content at 2–4 days (Figure 6C).

## 3. Discussion

Our findings indicate that Arg treatment inhibited the disease index and improved resistance to *A. tenuissima* in blueberries. Arg treatment was found to enhance the activities and gene transcription levels of GLU and CHI and activate ROS metabolism. In addition, Arg treatment promoted JA accumulation by upregulating gene expression involved in JA biosynthesis. 

ROS may function as signaling molecules that induce disease resistance-related responses [21]. Considering that the H_2_O_2_ content in the Arg-treated group increased initially during storage, it was indicated that Arg treatment resulted in an occurrence of oxygen stress (Figure 4). The same finding has been shown for blueberries treated with burdock fructooligosaccharides and cherry tomatoes treated with melatonin [22,23]. However, further evidence has also demonstrated that excessive ROS can adversely affect disease resistance [24,25,26]. Fruits and vegetables have a self-regulatory protection mechanism that activates the antioxidant defense system, thereby inhibiting excessive ROS accumulation and reducing oxidative damage [27,28]. This antioxidant defense system contains antioxidant enzymes, including APX, SOD, CAT, and POD, which can prevent the over-production of ROS [29]. At 0–0.5 d, the SOD activity in control group remained at low levels under the stress of pathogen. However, the SOD activity in Arg-treated fruit was significantly higher than in control, and Arg treatment induced the resistance of blueberry fruit to pathogen in this process (*p* < 0.05). At 0.5–2 d, the SOD activity in Arg-treated fruit was enhanced, which could be due to the antioxidant defense system in Arg-treated blueberry fruit was stimulated to defense pathogen. Through our study, we discovered that Arg treatment could increase CAT, SOD, POD, and APX activity and promote related gene expression in blueberries (Figure 5). Similar to our results, blueberries treated with thymol have been shown to exhibit higher SOD, CAT, and POD activity to control postharvest disease against *Aspergillus niger* [30].

GLU and CHI are PRs, which are hydrolases produced to improve plant resistance to foreign pathogenic bacteria [31]. Research has found that low-dose UV-C irradiation and melatonin treatment could enhance GLU and CHI enzyme activity and related gene expression, thereby promoting resistance to fungal decay [32,33]. Quercetin treatment has been proven to activate GLU and CHI and upregulate the related gene expression levels, thereby inducing resistance to *Penicillium expansum* in kiwifruit [34]. In our study, we determined Arg treatment improved GLU and CHI activity in blueberries as well as the gene expression of *VcGLU2* and *VcCHI*, thereby inducing resistance to *A. tenuissima* (Figure 2). In addition, polyphenolic compounds, as an abundant secondary metabolite in plants, are known to have a key role in inhibiting pathogenic growth [35]. Arg treatment was found to maintain higher content of total phenolics, flavonoids, and anthocyanin in postharvest blueberries, thereby improving disease resistance (Figure 6). 

The JA biosynthesis pathway acts an essential role in plant resistance to disease [36,37]. Plants activate the JA signaling pathway as a part of their defense responses, the possible mechanism of which is that JA can induce the transcription of related genes and promote secondary metabolism [38]. Endogenous JA content is involved in a plant’s resistance to a pathogen, which is induced by elicitor treatment [39]. In our study, Arg treatment was found to promote JA accumulation in blueberries. Similarly, in previous research, endogenous JA content was found to increase in peaches treated with sodium nitroprusside, thereby enhancing their resistance to *Monilinia fructicola* [40]. The biosynthesis of JA is mainly catalyzed by key enzymes (LOX, AOS, AOC, and OPR). When JA levels were low, the *JAZ* degradation rate reduced and the release of transcription factor *MYC2* was inhibited, thus inhibiting JA response in plants [41]. Methyl jasmonate (MeJA) was found to induce the relative expression of *PpLOX*, *PpOPR3*, and *PpAOS* in peaches to resist *Rhizopus stolonifera* [42]. MeJA also promotes key gene expression required for JA biosynthesis (*PavLOX*, *PavAOS*, *PavOPR3*, and *PavMYC2*) and improves resistance to *A. tenuissima* in sweet cherries [43]. Similarly, blueberries treated with melatonin have been found to induce resistance through the JA signaling pathway [44]. In the present study, Arg treatment could induce the gene expression levels in the blueberries involved in JA biosynthesis (*VcLOX1*, *VcAOS1*, *VcAOC*, *VcAOC3*, *VcOPR1*, *VcOPR3*, *VcMYC2*, and *VcCOI1*), ultimately activating the JA signaling pathway (Figure 3). Furthermore, the findings indicated that Arg treatment induced resistance to *A. tenuissima* in blueberries through triggering JA biosynthesis. Therefore, our findings provided evidence indicating that Arg treatment activated PRs and antioxidant enzymes, stimulated JA production, and further enhanced resistance to *A. tenuissima* in blueberries (Figure 7). 

In sum, these findings provide additional evidence indicating that Arg stimulates the production of JA, further enhancing resistance to *A. tenuissima* in blueberry fruit. In our upcoming study, we aim to explore the role of Arg in controlling the postharvest blueberry fruit against *A. tenuissima*. Our research will focus on the molecular biology aspect of the transcription factors (TFs), further exploring the regulation mechanism of Arg on disease resistance.

## 4. Materials and Methods

### 4.1. Pathogen

The strain B20190712E1, previously isolated from rotten blueberries, was identified as *A. tenuissima* [22]. It was cultured on potato dextrose agar at 25 °C for 15 days and a spore suspension of 1 × 10^5^ CFU mL^−1^ was prepared. 

### 4.2. Arg Treatment and Pathogen Inoculation

The blueberries (cv. Bluecrop) were picked in Shenyang, Liaoning, China (123°27′ E, 41°48′ N). Blueberries with the same size and level of maturity were selected for further testing.

A 2% (*w*/*v*) sodium hypochlorite solution was used to soak and disinfect the blueberries for three minutes. Samples were washed carefully using distilled water, dried, and divided into two groups, each containing 3 kg of blueberries. In the control group, the blueberries were sprayed with sterile water; for the Arg treatment group, the blueberries were sprayed with 1 mM Arg solution. Both groups were applied with the same amount. After 12 h, the spore suspension (10 μL) containing 1.0 × 10^5^ CFU mL^−1^ of *A. tenuissima* was inoculated at the pedicle. The experiments were repeated three times. The treated blueberries were subsequently stored at 20 °C and 80% RH, and samples were taken at 0, 0.25, 0.5, 1, 2, 3, 4, 5, and 6 days after inoculation. 

### 4.3. Assessment of Disease Index

The disease index of blueberry fruit was measured according to methods developed by Wang [45]. The percentage of the total damaged surface area in each blueberry in relation to the total epidermal area of the blueberry was graded. The scale was as follows: Scale 0, no visible rot on the fruit; scale 1, slight decay observed on the fruit and a decayed area lower than 25%; scale 2, moderate decay and a decayed area between 25% and 50%; scale 3, severe decay and a decayed area higher than 50%. The experiment was conducted three times. Each group contained approximately 120 blueberry fruits. The disease index calculation can be expressed as follows:Disease Index (%) = ∑(S × N)/(N_0_ × S_0_) × 100%(1)
where S represents the disease scale, N represents the number of fruits of the corresponding scale, N_0_ represents the total quantity of fruits, and S_0_ represents the maximum scale.

### 4.4. Assessment of JA Content

An ELISA kit was used to measure JA content according to the instructions (Jiangsu Meimian Industrial, Yancheng, China).

### 4.5. Measurement of GLU and CHI Activity

ELISA kits (Jiangsu Baolai Technology, Yancheng, China) were utilized to determine GLU and CHI activity, with reference to Shu [46] for modifications.

### 4.6. Measurement of ROS Levels and Key Antioxidant Enzyme Activities

#### 4.6.1. Determination of H_2_O_2_ and MDA Levels

H_2_O_2_ content was determined by methods developed by Yan [47], with a few changes. First, 3.0 g of blueberry tissue was weighed and homogenized with 5.0 mL of pre-cooled acetone, and then centrifuged (12,000× *g* at 4 °C for 20 min). Subsequently, the precipitate was washed with pre-cooled acetone, dissolved in sulfuric acid solution (3 mL 2 mol L^−1^), and examined at 412 nm. 

To measure MDA content, we followed the methods developed by Ma [48]. We homogenized 3.0 g of blueberry tissue in 10% (*w*/*v*) trichloroacetic acid (TCA). This mixture was subsequently centrifuged (10,000× *g* at 4 °C for 20 min). The supernatant was collected and its absorbance was determined at 450, 532, and 600 nm.

#### 4.6.2. Analysis of Antioxidant Enzyme Activity

The method developed by Ge [49] was used to measure SOD activity with some modifications. One SOD activity unit was determined to be the quantity of enzyme that inhibited 50% of the photochemical reduction of nitroblue tetrazolium (NBT), estimated at 560 nm.

The method developed by Wang [50] was utilized with some changes for the assessment of CAT activity. Approximately 3.0 g blueberry tissue was added to 5 mL of extraction buffer, which contained 5 mmol L^−1^ dithiothreitol (DTT) and 5% polyvinylpolypyrrolidone. The mixture underwent homogenization and was centrifuged at 4 °C and 12,000× *g* for 30 min. A CAT enzymatic reaction system of 0.01 reduction per gram in one minute was determined to be one unit of CAT activity. 

The APX activity was assayed using methods described by Ren [51], with some changes. First, 3.0 g blueberry fruit tissue was ground in 5 mL extraction buffer, which contained 0.1 mmol L^−1^ EDTA, 1 mmol L^−1^ ascorbic acid, and 2% PVPP. Then, the mixture was centrifuged. One unit of APX was determined to be the required quantity to achieve a 0.01 reduction in the fruit sample (290 nm) every minute.

POD activity was assayed with slight modifications based on methods developed by Alijani [52]. The extraction buffer contained 4% crosslinked polyvinyl-poly-pyrrolidone (PVPP) (*w*/*v*), 1 mmol polyethylene glycol 6000 (PEG 6000), and 1% Triton X-100 (*v/v*). A total of 3.0 g of fresh blueberry tissue was homogenized in extraction buffer (5 mL) and centrifuged. One unit of POD activity was defined as the amount of POD required to enhance the absorbance (470 nm) by 1 nm in one minute. 

### 4.7. Measurement of Total Phenolic, Flavonoid, and Anthocyanin Content

Phenolic, flavonoid, and anthocyanin content was evaluated using the method presented by Wang [53], with some changes. The mass of gallic acid equivalents was used to characterize the total phenolic content measured at 280 nm. Flavonoid and anthocyanin content were expressed as the mass of rutin and cyanidin-3-glucoside equivalents, respectively. The absorbance was determined at 280, 325, 600, and 530 nm.

### 4.8. Real-Time Quantitative PCR (RT-qPCR)

Several differentially expressed genes involved in ROS, PRs, and JA biosynthesis pathways were identified by transcriptomic analysis of inoculated control and Arg-treated blueberries. Primer 5.0 (Premier Biosoft, Palo Alto, CA, USA) was used to design gene primers (Appendix A Table A1) and the primers were synthesized by Sangon (Shanghai, China). RT-qPCR amplification and analysis were conducted by the UltraSYBR Master Mix Kit (CWBIO, Beijing, China). The relative gene expression was computed by the 2^−∆∆CT^ method. For internal reference, we used the glyceraldehyde-3-phosphate dehydrogenase (*GAPDH*) gene [54].

### 4.9. Statistical Analysis

The experiments were repeated three times. In this study, the *t*-test was performed to analyze data using SPSS Statistics 25, and *p*-values where *p* < 0.05 were considered to be statistically significant. In addition, GraphPad Prism 8 (GraphPad Software Inc., La Jolla, CA, USA) was used for data analysis and graphics.

## 5. Conclusions

Arg treatment was found to induce resistance to *A. tenuissima* in postharvest blueberries. In particular, Arg was observed to trigger an initial increase in H_2_O_2_, leading to an early defense response in the Arg-treated blueberries. Arg treatment was found to protect the blueberry fruit from oxidative damage. Arg also induced resistance to *A. tenuissima* by facilitating the gene expression of GLU, CHI, and antioxidant enzymes. Furthermore, Arg treatment stimulated JA production by inducing the expression of key genes in JA biosynthesis, thus triggering the defense response. Therefore, Arg treatment may be a promising and effective method to control *Alternaria* fruit rot in postharvest blueberries.

## Figures and Tables

**Figure 1 plants-13-01058-f001:**
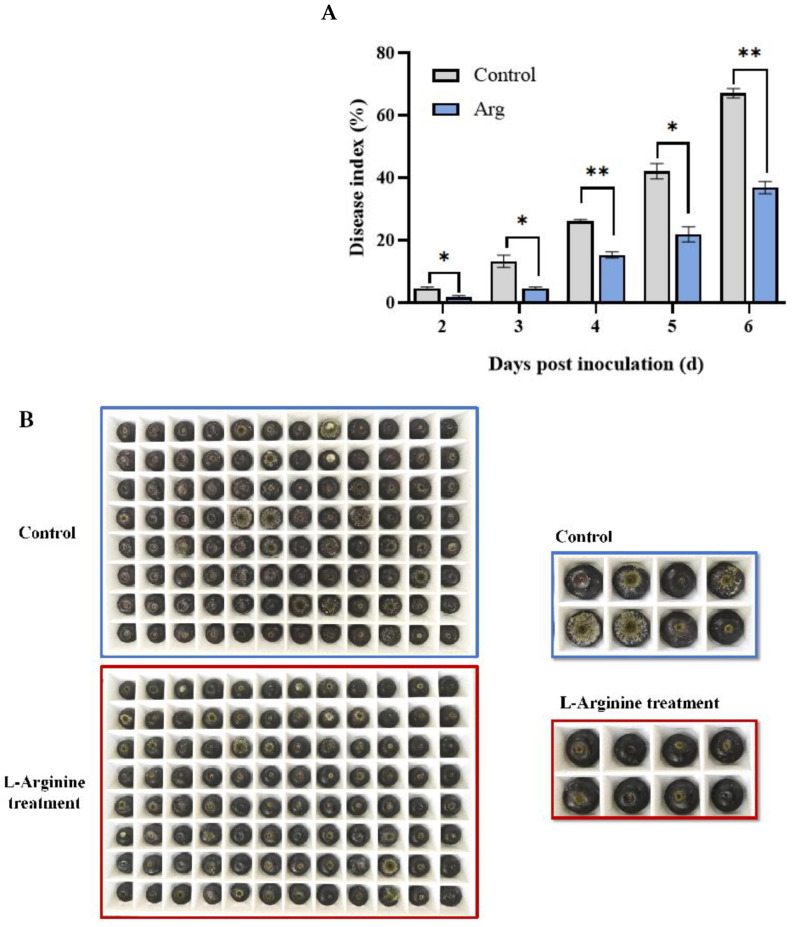
Disease index (**A**) in control and L-arginine (Arg) treatment of inoculated blueberries. Symptoms of control and Arg-inoculated blueberry fruit at 6 d (**B**). The data and bar represent three repeating averages and standard deviations. The asterisk indicates a significant difference between the control and Arg groups based on the independent sample *t*-test (* *p* < 0.05, ** *p* < 0.01).

**Figure 2 plants-13-01058-f002:**
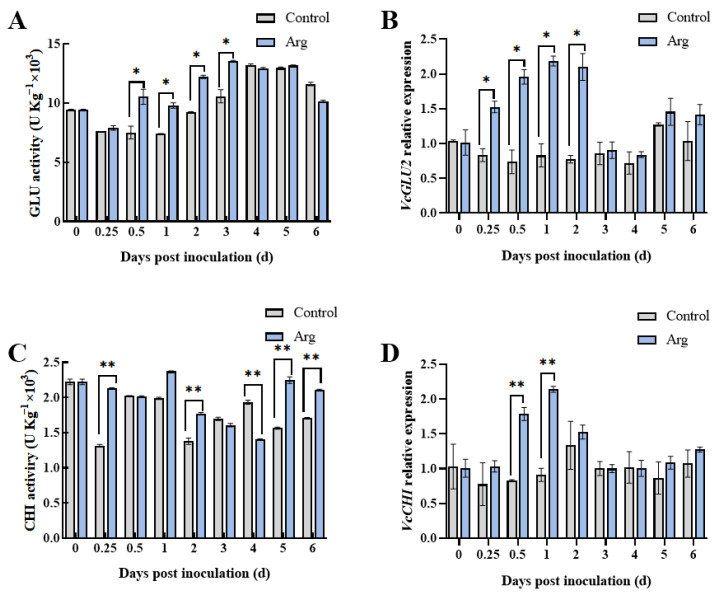
Changes in GLU (**A**) activity and *VcGLU2* relative expression (**B**). The activity of chitinase (CHI) (**C**) and relative expression of *VcCHI* (**D**). The data and bar represent three repeating averages and standard deviations. * *p* < 0.05, ** *p* < 0.01.

**Figure 3 plants-13-01058-f003:**
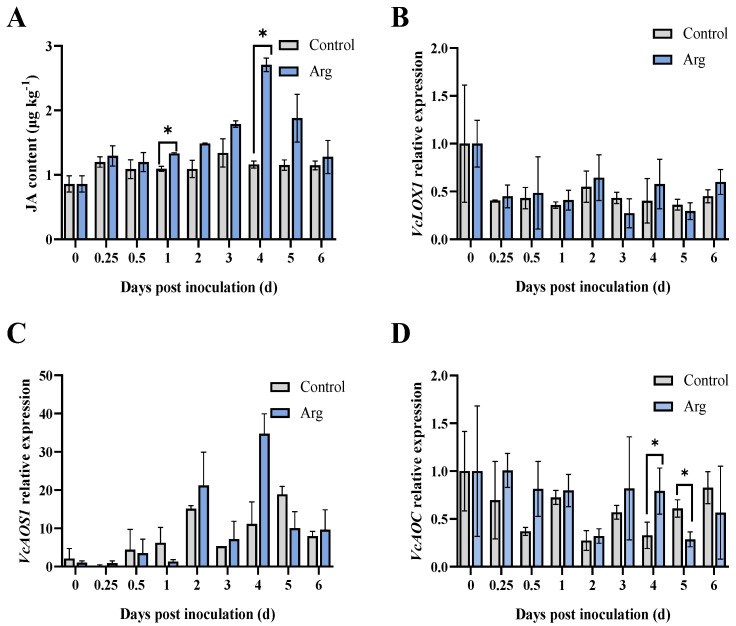
Changes in JA content (**A**), and the relative gene expression levels of *VcLOX1* (**B**), *VcAOS1* (**C**), *VcAOC* (**D**), *VcAOC3* (**E**), *VcOPR1* (**F**), *VcOPR3* (**G**), *VcMYC2* (**H**), and *VcCOI1* (**I**) of control and L-arginine (Arg) treatment in inoculated blueberry fruit groups. The data and bars represent the average values and standard deviation of three replicates. * *p* < 0.05, ** *p* < 0.01.

**Figure 4 plants-13-01058-f004:**
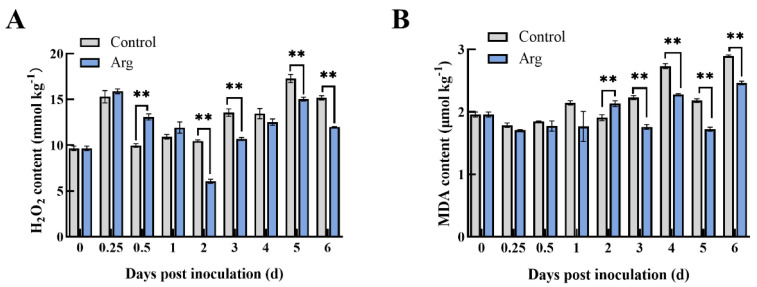
The content of H_2_O_2_ (**A**) and MDA (**B**) of inoculated blueberry fruit in control and L-arginine (Arg) group. The data are expressed as three repeated mean ± standard deviation. ** *p* < 0.01.

**Figure 5 plants-13-01058-f005:**
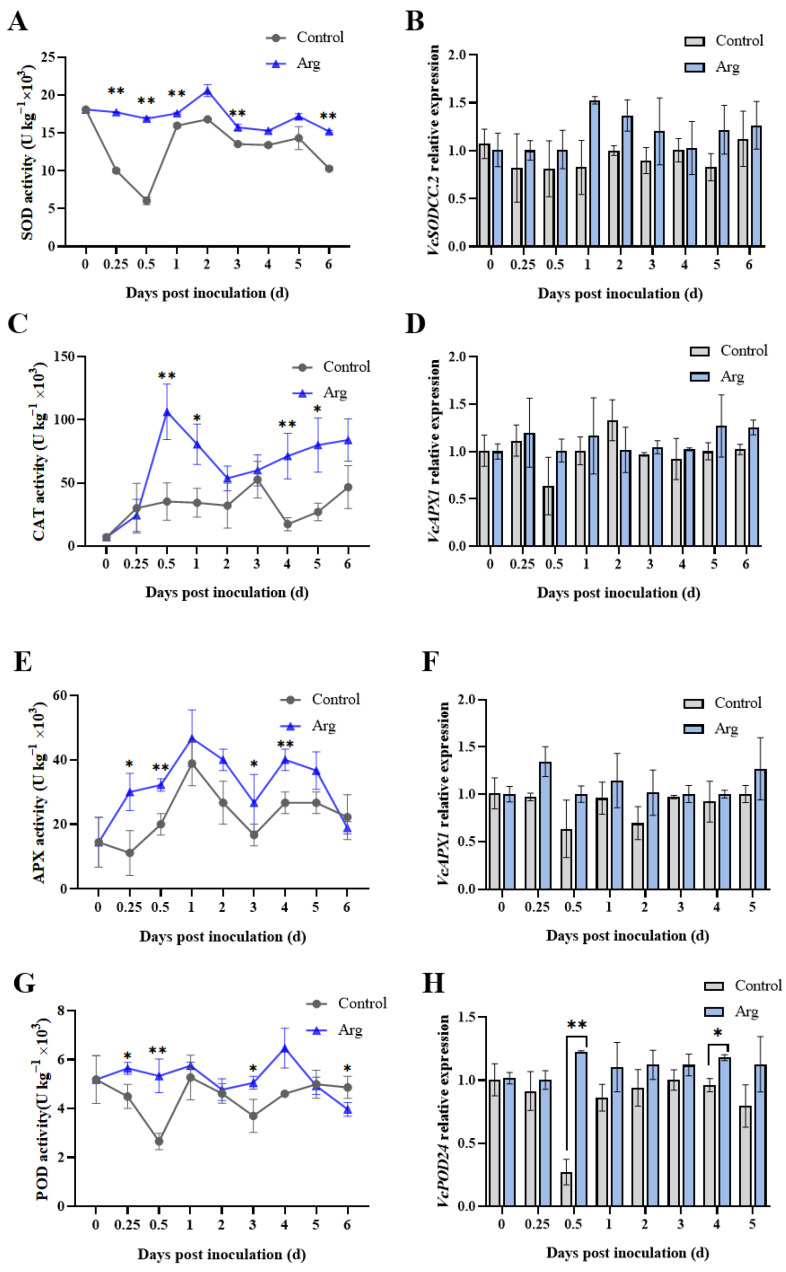
The activities of SOD (**A**), CAT (**C**), APX (**E**), and POD (**G**). The relative expression of *VcSODCC.2* (**B**), *VcCAT2* (**D**), *VcAPX1* (**F**), and *VcPOD24* (**H**). The data and bars represent three repeating averages and standard deviations. * *p* < 0.05; ** *p* < 0.01.

**Figure 6 plants-13-01058-f006:**
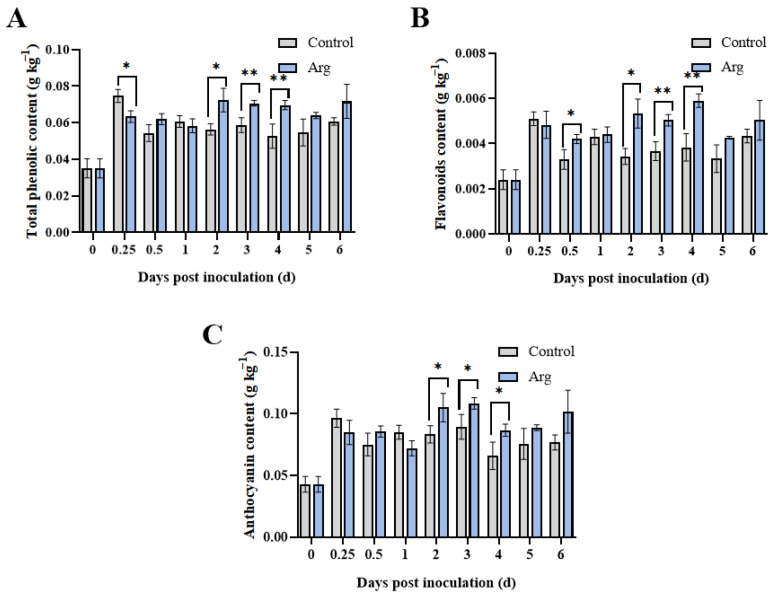
Content of total phenolics (**A**), flavonoids (**B**), and anthocyanin (**C**) of control and L-arginine (Arg) treatment in inoculated blueberry fruit. * *p* < 0.05, ** *p* < 0.01.

**Figure 7 plants-13-01058-f007:**
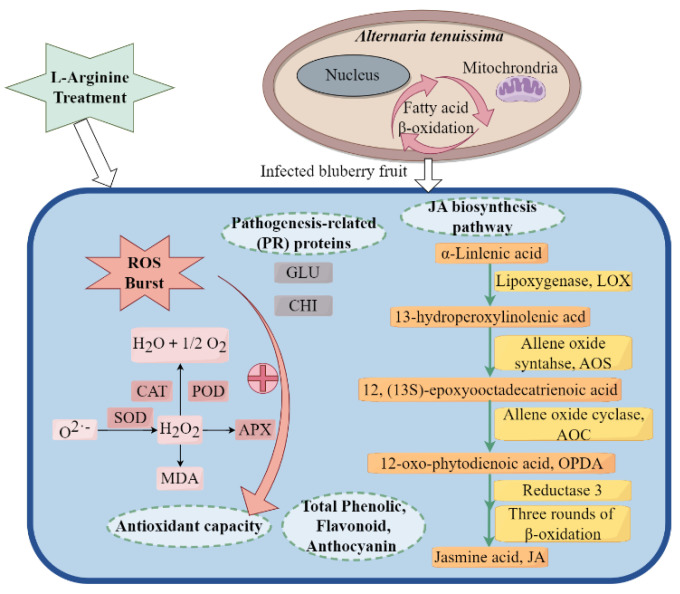
Supposed L-arginine (Arg) mechanism that induces resistance against *A. tenuissima* in postharvest blueberry fruit. This figure was drawn by Figdraw.

## Data Availability

Data are available from the authors upon request. The data are not publicly available due to privacy.

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
