# Peer review of "Effect and Mechanism of L-Arginine against Alternaria Fruit Rot in Postharvest Blueberry Fruit"

_plants, 2024, doi:10.3390/plants13081058_

Round 1

Reviewer 1 Report

Comments and Suggestions for Authors

Summary

In this paper, the authors explore the impact of L-Arginine (Arg) on the development of resistance to the Alternaria tenuissima pathogen in blueberries. For this purpose, the ROS metabolism, pathogenesis-related proteins (PRs), and jasmonic acid (JA) biosynthesis pathways were analyzed, including changes in activity and gene expression of key enzymes. The authors indicated that Arg treatment prevented the postharvest development of Alternaria-caused rot in blueberry fruits. The authors also found that Arg treatment induces a burst of hydrogen peroxide in the blueberries early on during storage, thereby improving their resistance to Alternaria tenuissima. The authors have also shown that Arg treatment increased the activity of AOX enzymes (POX, CAT, SOD, and APX) and related gene expression, as well as the total levels of phenolics, flavonoids, and anthocyanin. Also, the activity and gene expression of the PRs were elevated in Arg-treated blueberries, boosting their resistance to pathogens. The increase in endogenous JA content was detected in Arg-treated blueberries, along with upregulated expression of key genes related to the JA biosynthesis pathway, further boosting disease resistance. In the end, the authors concluded that Arg treatment was determined to be a promising and prospective method for controlling Alternaria fruit rot in blueberries.

Specific remarks

Line 17: It's unclear what the sentence says. One ought to think about rearranging the statement.

Lines 14, 17, 19, 28, etc.: Please be consistent with naming the pathogen: Alternaria tenuissima, Alternaria, and A. tenuissima.

Figure 1B: The quality of the picture is very poor, which disables the reader from following up on the experimental results.

Discussion: We can observe the significant drop in the SOD activity in control samples after 0.25 and 0.5 and the increase after 1 day, pointing to the excessive change of ROS-related enzyme activity without the influence of Arg. However, this is not the case with Arg-treated samples. If Arg induces oxidative stress, it is strange that a similar SOD trend does not occur with Arg-treated samples. Authors should try to comment on these results in discussion.

Comments on the Quality of English Language

The quality of the English language is fine.

Author Response

On behalf of my co-authors, we thank you very much for allowing us to revise our manuscript. We are truly grateful for these critical comments and thoughtful suggestions on our manuscript entitled “Effect and Mechanism of L-Arginine against Alternaria Fruit Rot in Postharvest Blueberry fruit” (plants-2928958). These comments are all valuable and very helpful for revising and improving our paper, as well as the important guiding significance to our research. Based on these comments and suggestions, we have made careful revisions to the original manuscript and provided a point-by-point response to every question. Revised portions are marked in red in the manuscript. The main corrections in the paper and the responses to the comments are as follows:

Reviewer: 1

Specific remarks

  • Line 17: It's unclear what the sentence says. One ought to think about rearranging the statement.

Reply: Thank you for your professional and patient suggestions. We apologized for the mistake in this sentence. We have rearranged the sentence. Please refer to lines 18-20.

  • Lines 14, 17, 19, 28, etc.: Please be consistent with naming the pathogen: Alternaria tenuissima, Alternaria, and tenuissima.

Reply: Thank you for your professional and patient suggestion. “A. tenuissima” is the shorthand of “Alternaria tenuissima”. We revised and explained it when “Alternaria tenuissima first appeared in the manuscript and please refer to line 16. Alternaria is a genus of which Alternaria tenuissima is a species.

  • Figure 1B: The quality of the picture is very poor, which disables the reader from following up on the experimental results.

Reply: Thank you for your professional suggestions on improving our manuscript; we have revised it. Please refer to Figure 1B.

  • Discussion: We can observe the significant drop in the SOD activity in control samples after 0.25 and 0.5 and the increase after 1 day, pointing to the excessive change of ROS-related enzyme activity without the influence of Arg. However, this is not the case with Arg-treated samples. If Arg induces oxidative stress, it is strange that a similar SOD trend does not occur with Arg-treated samples. Authors should try to comment on these results in discussion.

Reply: Thank you for your professional suggestions on improving our manuscript; we have added some analyses. Please refer to lines 221-226.

We have tried our best to improve the manuscript and made some changes in the manuscript. We are grateful for your professional comments on our paper. We hope that the correction will meet with your approval. 

Once again, thank you very much for your comments and suggestions.

Yours Sincerely,

Jiaqi Wang

* Correspondence should be addressed to Prof. Xuerui Yan, College of Food Science, Shenyang Agricultural University, Shenyang, 110866, PR. China. Tel: 024-88487161

Reviewer 2 Report

Comments and Suggestions for Authors

In my opinion it is an interesting well written article. I have only several minor comments:

1/38 and elswhere Alternaria fruit rot is a common disease name, thus Alternaria should not be in italics.

1/74 If the headline is in italics, then the scientific name (A. tenuissima) should be written normally.

Introduction: I would recommend to add an info which Alternaria species are able to cause blueberry fruit rot, and what is the position of A. tenuissima among them (more or less important).

2/93 rewrite the sentence ...blueberries was significantly higher...

10/229 in my opinion: Arg treatment was found to maintain the higher content

10/235 and elswhere not resistance to the disease, but to the pathogen

10/238 ...TO to Monilinia fructicola.

Author Response

On behalf of my co-authors, we thank you very much for allowing us to revise our manuscript. We are truly grateful for these critical comments and thoughtful suggestions on our manuscript entitled “Effect and Mechanism of L-Arginine against Alternaria Fruit Rot in Postharvest Blueberry fruit” (plants-2928958). These comments are all valuable and very helpful for revising and improving our paper, as well as the important guiding significance to our research. Based on these comments and suggestions, we have made careful revisions to the original manuscript and provided a point-by-point response to every question. Revised portions are marked in red in the manuscript. The main corrections in the paper and the responses to the comments are as follows:

Reviewer: 2

Comments and Suggestions for Authors

In my opinion it is an interesting well written article. I have only several minor comments:

(1) 1/38 and elswhere Alternaria fruit rot is a common disease name, thus Alternaria should not be in italics.

Reply: Thank you for your professional suggestions on improving our manuscript; we have revised it. Please refer to the headline and lines 2, 19,30, 42 and 49-49.

(2) 1/74 If the headline is in italics, then the scientific name (A. tenuissima) should be written normally.

Reply: Thank you for your professional and patient suggestion. Based on your above comment, I really agree with your opinion. We have revised “Alternaria” in the headline to be written normally. In addtion, “A. tenuissima” is one of the species in Alternaria sp., so we used it in italics. Please refer to line 16.

(3) Introduction: I would recommend to add an info which Alternariaspecies are able to cause blueberry fruit rot, and what is the position of tenuissima among them (more or less important).

Reply: Thank you for your professional suggestions on improving our manuscript. We have revised it and added an info which Alternaria species are able to cause blueberry fruit rot, and what is the position of A. tenuissima among them. Please refer to lines 40-45.

(4) 2/93 rewrite the sentence ...blueberrieswas significantly higher...

Reply: Thank you for your professional and patient suggestion. We sincerely apologize for the occurrence of such a problem. Thank you for pointing this out and we have improved it. We have checked carefully to ensure that there were no mistakes and revised them. Please refer to line 102.

(5) 10/229 in my opinion: Arg treatment was found to maintain the higher content

Reply: Thank you for your professional suggestions on improving our manuscript. We have revised the sentence. Please refer to line 241.

(6) 10/235 and elswhere not resistance to the disease, but to the pathogen

Reply: Thank you for your professional and patient suggestion. We have revised “disease” to “the pathogen” to make the sentence clearer. Please refer to line 247.

(7) 10/238 ...TO to Monilinia fructicola.

Reply: Thank you for your professional and patient suggestion. We sincerely apologize for the occurrence of such a problem. Thank you for pointing this out and we have improved it. We have checked carefully to ensure that there was no mistake and revised it. Please refer to line 250.

We have tried our best to improve the manuscript and made some changes in the manuscript. We are grateful for your professional comments on our paper. We hope that the correction will meet with your approval. 

Once again, thank you very much for your comments and suggestions.

Yours Sincerely,

Jiaqi Wang

* Correspondence should be addressed to Prof. Xuerui Yan, College of Food Science, Shenyang Agricultural University, Shenyang, 110866, PR. China. Tel: 024-88487161
